# Recurrent Models of Visual Attention

**Volodymyr Mnih    Nicolas Heess    Alex Graves    Koray Kavukcuoglu**
Google DeepMind

{vmnih,heess,gravesa,korayk} @ google.com

## Abstract

Applying convolutional neural networks to large images is computationally expensive because the amount of computation scales linearly with the number of image pixels. We present a novel recurrent neural network model that is capable of extracting information from an image or video by adaptively selecting a sequence of regions or locations and only processing the selected regions at high resolution. Like convolutional neural networks, the proposed model has a degree of translation invariance built-in, but the amount of computation it performs can be controlled independently of the input image size. While the model is non-differentiable, it can be trained using reinforcement learning methods to learn task-specific policies. We evaluate our model on several image classification tasks, where it significantly outperforms a convolutional neural network baseline on cluttered images, and on a dynamic visual control problem, where it learns to track a simple object without an explicit training signal for doing so.

## 1   Introduction

Neural network-based architectures have recently had great success in significantly advancing the state of the art on challenging image classification and object detection datasets [8, 12, 19]. Their excellent recognition accuracy, however, comes at a high computational cost both at training and testing time. The large convolutional neural networks typically used currently take days to train on multiple GPUs even though the input images are downsampled to reduce computation [12]. In the case of object detection processing a single image at test time currently takes seconds when running on a single GPU [8, 19] as these approaches effectively follow the classical sliding window paradigm from the computer vision literature where a classifier, trained to detect an object in a tightly cropped bounding box, is applied independently to thousands of candidate windows from the test image at different positions and scales. Although some computations can be shared, the main computational expense for these models comes from convolving filter maps with the entire input image, therefore their computational complexity is at least linear in the number of pixels.

One important property of human perception is that one does not tend to process a whole scene in its entirety at once. Instead humans focus attention selectively on parts of the visual space to acquire information when and where it is needed, and combine information from different fixations over time to build up an internal representation of the scene [18], guiding future eye movements and decision making. Focusing the computational resources on parts of a scene saves "bandwidth" as fewer "pixels" need to be processed. But it also substantially reduces the task complexity as the object of interest can be placed in the center of the fixation and irrelevant features of the visual environment ("clutter") outside the fixated region are naturally ignored.

In line with its fundamental role, the guidance of human eye movements has been extensively studied in neuroscience and cognitive science literature. While low-level scene properties and bottom up processes (e.g. in the form of saliency; [11]) play an important role, the locations on which humans fixate have also been shown to be strongly task specific (see [9] for a review and also e.g. [15, 22]). In this paper we take inspiration from these results and develop a novel framework for attention-based task-driven visual processing with neural networks. Our model considers attention-based processing

of a visual scene as a *control problem* and is general enough to be applied to static images, videos, or as a perceptual module of an agent that interacts with a dynamic visual environment (e.g. robots, computer game playing agents).

The model is a recurrent neural network (RNN) which processes inputs sequentially, attending to different locations within the images (or video frames) one at a time, and incrementally combines information from these fixations to build up a dynamic internal representation of the scene or environment. Instead of processing an entire image or even bounding box at once, at each step, the model selects the next location to attend to based on past information *and* the demands of the task. Both the number of parameters in our model and the amount of computation it performs can be controlled independently of the size of the input image, which is in contrast to convolutional networks whose computational demands scale linearly with the number of image pixels. We describe an end-to-end optimization procedure that allows the model to be trained directly with respect to a given task and to maximize a performance measure which may depend on the entire sequence of decisions made by the model. This procedure uses backpropagation to train the neural-network components and policy gradient to address the non-differentiabilities due to the control problem.

We show that our model can learn effective task-specific strategies for where to look on several image classification tasks as well as a dynamic visual control problem. Our results also suggest that an attention-based model may be better than a convolutional neural network at both dealing with clutter and scaling up to large input images.

## 2 Previous Work

Computational limitations have received much attention in the computer vision literature. For instance, for object detection, much work has been dedicated to reducing the cost of the widespread sliding window paradigm, focusing primarily on reducing the number of windows for which the full classifier is evaluated, e.g. via classifier cascades (e.g. [7, 24]), removing image regions from consideration via a branch and bound approach on the classifier output (e.g. [13]), or by proposing candidate windows that are likely to contain objects (e.g. [1, 23]). Even though substantial speedups may be obtained with such approaches, and some of these can be combined with or used as an add-on to CNN classifiers [8], they remain firmly rooted in the window classifier design for object detection and only exploit past information to inform future processing of the image in a very limited way.

A second class of approaches that has a long history in computer vision and is strongly motivated by human perception are saliency detectors (e.g. [11]). These approaches prioritize the processing of potentially interesting ("salient") image regions which are typically identified based on some measure of local low-level feature contrast. Saliency detectors indeed capture some of the properties of human eye movements, but they typically do not to integrate information across fixations, their saliency computations are mostly hardwired, and they are based on low-level image properties only, usually ignoring other factors such as semantic content of a scene and task demands (but see [22]).

Some works in the computer vision literature and elsewhere e.g. [2, 4, 6, 14, 16, 17, 20] have embraced vision as a sequential decision task as we do here. There, as in our work, information about the image is gathered sequentially and the decision where to attend next is based on previous fixations of the image. [4] employs the learned Bayesian observer model from [5] to the task of object detection. The learning framework of [5] is related to ours as they also employ a policy gradient formulation (cf. section 3) but their overall setup is considerably more restrictive than ours and only some parts of the system are learned.

Our work is perhaps the most similar to the other attempts to implement attentional processing in a deep learning framework [6, 14, 17]. Our formulation which employs an RNN to integrate visual information over time and to decide how to act is, however, more general, and our learning procedure allows for end-to-end optimization of the sequential decision process instead of relying on greedy action selection. We further demonstrate how the same general architecture can be used for efficient object recognition in still images as well as to interact with a dynamic visual environment in a task-driven way.

## 3 The Recurrent Attention Model (RAM)

In this paper we consider the attention problem as the sequential decision process of a goal-directed agent interacting with a visual environment. At each point in time, the agent observes the environment only via a bandwidth-limited sensor, i.e. it never senses the environment in full. It may extract

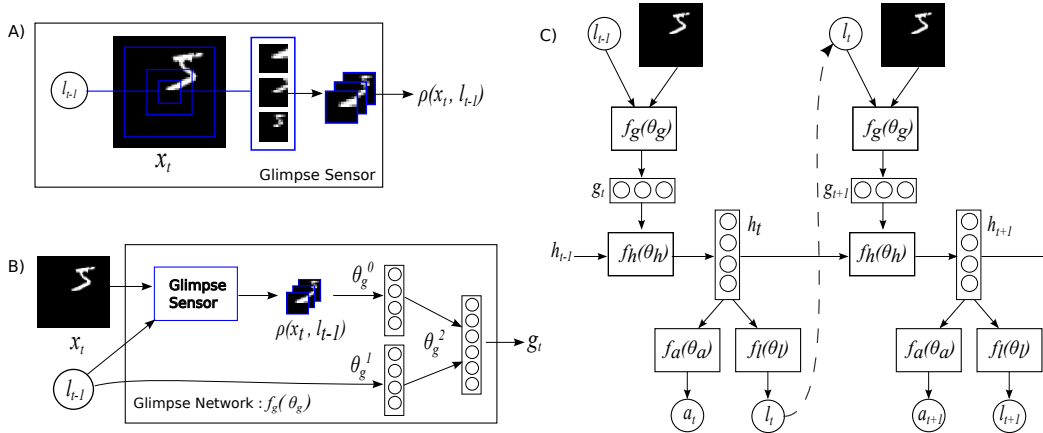

Figure 1: **A) Glimpse Sensor:** Given the coordinates of the glimpse and an input image, the sensor extracts a *retina-like* representation $\rho(x_t, l_{t-1})$ centered at $l_{t-1}$ that contains multiple resolution patches. **B) Glimpse Network:** Given the location $(l_{t-1})$ and input image $(x_t)$, uses the glimpse sensor to extract retina representation $\rho(x_t, l_{t-1})$. The retina representation and glimpse location is then mapped into a hidden space using independent linear layers parameterized by $\theta_g^0$ and $\theta_g^1$ respectively using rectified units followed by another linear layer $\theta_g^2$ to combine the information from both components. The glimpse network $f_g(.; \{\theta_g^0, \theta_g^1, \theta_g^2\})$ defines a trainable bandwidth limited sensor for the attention network producing the glimpse representation $g_t$. **C) Model Architecture:** Overall, the model is an RNN. The core network of the model $f_h(.; \theta_h)$ takes the glimpse representation $g_t$ as input and combining with the internal representation at previous time step $h_{t-1}$, produces the new internal state of the model $h_t$. The location network $f_l(.; \theta_l)$ and the action network $f_a(.; \theta_a)$ use the internal state $h_t$ of the model to produce the next location to attend to $l_t$ and the action/classification $a_t$ respectively. This basic RNN iteration is repeated for a variable number of steps.

information only in a local region or in a narrow frequency band. The agent can, however, actively control how to deploy its sensor resources (e.g. choose the sensor location). The agent can also affect the true state of the environment by executing actions. Since the environment is only partially observed the agent needs to integrate information over time in order to determine how to act and how to deploy its sensor most effectively. At each step, the agent receives a scalar reward (which depends on the actions the agent has executed and can be delayed), and the goal of the agent is to maximize the total sum of such rewards.

This formulation encompasses tasks as diverse as object detection in static images and control problems like playing a computer game from the image stream visible on the screen. For a game, the environment state would be the true state of the game engine and the agent's sensor would operate on the video frame shown on the screen. (Note that for most games, a single frame would not fully specify the game state). The environment actions here would correspond to joystick controls, and the reward would reflect points scored. For object detection in static images the state of the environment would be fixed and correspond to the true contents of the image. The environmental action would correspond to the classification decision (which may be executed only after a fixed number of fixations), and the reward would reflect if the decision is correct.

### 3.1 Model

The agent is built around a recurrent neural network as shown in Fig. 1. At each time step, it processes the sensor data, integrates information over time, and chooses how to act and how to deploy its sensor at next time step:

**Sensor:** At each step $t$ the agent receives a (partial) observation of the environment in the form of an image $x_t$. The agent does not have full access to this image but rather can extract information from $x_t$ via its bandwidth limited sensor $\rho$, e.g. by focusing the sensor on some region or frequency band of interest.

In this paper we assume that the bandwidth-limited sensor extracts a retina-like representation $\rho(x_t, l_{t-1})$ around location $l_{t-1}$ from image $x_t$. It encodes the region around $l$ at a high-resolution but uses a progressively lower resolution for pixels further from $l$, resulting in a vector of much

lower dimensionality than the original image $x$. We will refer to this low-resolution representation as a *glimpse* [14]. The glimpse sensor is used inside what we call the *glimpse network $f_g$* to produce the glimpse feature vector $g_t = f_g(x_t, l_{t-1}; \theta_g)$ where $\theta_g = \{\theta_g^0, \theta_g^1, \theta_g^2\}$ (Fig. 1B).

**Internal state:** The agent maintains an interal state which summarizes information extracted from the history of past observations; it encodes the agent's knowledge of the environment and is instrumental to deciding how to act and where to deploy the sensor. This internal state is formed by the hidden units $h_t$ of the recurrent neural network and updated over time by the *core network*: $h_t = f_h(h_{t-1}, g_t; \theta_h)$. The external input to the network is the glimpse feature vector $g_t$.

**Actions:** At each step, the agent performs two actions: it decides how to deploy its sensor via the sensor control $l_t$, and an environment action $a_t$ which might affect the state of the environment. The nature of the environment action depends on the task. In this work, the location actions are chosen stochastically from a distribution parameterized by the location network $f_l(h_t; \theta_l)$ at time $t$: $l_t \sim p(\cdot | f_l(h_t; \theta_l))$. The environment action $a_t$ is similarly drawn from a distribution conditioned on a second network output $a_t \sim p(\cdot | f_a(h_t; \theta_a))$. For classification it is formulated using a softmax output and for dynamic environments, its exact formulation depends on the action set defined for that particular environment (e.g. joystick movements, motor control, ...). Finally, our model can also be augmented with an additional action that decides when it will stop taking glimpses. This could, for example, be used to learn a cost-sensitive classifier by giving the agent a negative reward for each glimpse it takes, forcing it to trade off making correct classifications with the cost of taking more glimpses.

**Reward:** After executing an action the agent receives a new visual observation of the environment $x_{t+1}$ and a reward signal $r_{t+1}$. The goal of the agent is to maximize the sum of the reward signal[1] which is usually very sparse and delayed: $R = \sum_{t=1}^{T} r_t$. In the case of object recognition, for example, $r_T = 1$ if the object is classified correctly after $T$ steps and 0 otherwise.

The above setup is a special instance of what is known in the RL community as a Partially Observable Markov Decision Process (POMDP). The true state of the environment (which can be static or dynamic) is unobserved. In this view, the agent needs to learn a (stochastic) policy $\pi((l_t, a_t)|s_{1:t}; \theta)$ with parameters $\theta$ that, at each step $t$, maps the history of past interactions with the environment $s_{1:t} = x_1, l_1, a_1, \ldots x_{t-1}, l_{t-1}, a_{t-1}, x_t$ to a distribution over actions for the current time step, subject to the constraint of the sensor. In our case, the policy $\pi$ is defined by the RNN outlined above, and the history $s_t$ is summarized in the state of the hidden units $h_t$. We will describe the specific choices for the above components in Section 4.

### 3.2 Training

The parameters of our agent are given by the parameters of the glimpse network, the core network (Fig. 1C), and the action network $\theta = \{\theta_g, \theta_h, \theta_a\}$ and we learn these to maximize the total reward the agent can expect when interacting with the environment.

More formally, the policy of the agent, possibly in combination with the dynamics of the environment (e.g. for game-playing), induces a distribution over possible interaction sequences $s_{1:N}$ and we aim to maximize the reward under this distribution: $J(\theta) = \mathbb{E}_{p(s_{1:T};\theta)} \left[ \sum_{t=1}^{T} r_t \right] = \mathbb{E}_{p(s_{1:T};\theta)} [R]$, where $p(s_{1:T}; \theta)$ depends on the policy

Maximizing $J$ exactly is non-trivial since it involves an expectation over the high-dimensional interaction sequences which may in turn involve unknown environment dynamics. Viewing the problem as a POMDP, however, allows us to bring techniques from the RL literature to bear: As shown by Williams [26] a sample approximation to the gradient is given by

$$\nabla_\theta J = \sum_{t=1}^{T} \mathbb{E}_{p(s_{1:T};\theta)} \left[ \nabla_\theta \log \pi(u_t | s_{1:t}; \theta) R \right] \approx \frac{1}{M} \sum_{i=1}^{M} \sum_{t=1}^{T} \nabla_\theta \log \pi(u_t^i | s_{1:t}^i; \theta) R^i, \quad (1)$$

where $s^i$'s are interaction sequences obtained by running the current agent $\pi_\theta$ for $i = 1 \ldots M$ episodes.

The learning rule (1) is also known as the REINFORCE rule, and it involves running the agent with its current policy to obtain samples of interaction sequences $s_{1:T}$ and then adjusting the parameters $\theta$ of our agent such that the log-probability of chosen actions that have led to high cumulative reward is increased, while that of actions having produced low reward is decreased.

Eq. (1) requires us to compute $\nabla_\theta \log \pi(u_t^i|s_{1:t}^i;\theta)$. But this is just the gradient of the RNN that defines our agent evaluated at time step $t$ and can be computed by standard backpropagation [25].

**Variance Reduction :** Equation (1) provides us with an unbiased estimate of the gradient but it may have high variance. It is therefore common to consider a gradient estimate of the form

$$\frac{1}{M}\sum_{i=1}^{M}\sum_{t=1}^{T}\nabla_\theta \log \pi(u_t^i|s_{1:t}^i;\theta)\left(R_t^i - b_t\right), \tag{2}$$

where $R_t^i = \sum_{t'=1}^{T} r_{t'}^i$ is the cumulative reward obtained *following* the execution of action $u_t^i$, and $b_t$ is a baseline that may depend on $s_{1:t}^i$ (e.g. via $h_t^i$) but not on the action $u_t^i$ itself. This estimate is equal to (1) in expectation but may have lower variance. It is natural to select $b_t = \mathbb{E}_\pi\left[R_t\right]$ [21], and this form of baseline known as the value function in the reinforcement learning literature. The resulting algorithm increases the log-probability of an action that was followed by a larger than expected cumulative reward, and decreases the probability if the obtained cumulative reward was smaller. We use this type of baseline and learn it by reducing the squared error between $R_t^i$'s and $b_t$.

**Using a Hybrid Supervised Loss:** The algorithm described above allows us to train the agent when the "best" actions are unknown, and the learning signal is only provided via the reward. For instance, we may not know a priori which sequence of fixations provides most information about an unknown image, but the total reward at the end of an episode will give us an indication whether the tried sequence was good or bad.

However, in some situations we do know the correct action to take: For instance, in an object detection task the agent has to output the label of the object as the final action. For the training images this label will be known and we can directly optimize the policy to output the correct label associated with a training image at the end of an observation sequence. This can be achieved, as is common in supervised learning, by maximizing the conditional probability of the true label given the observations from the image, i.e. by maximizing $\log \pi(a_T^*|s_{1:T};\theta)$, where $a_T^*$ corresponds to the ground-truth label(-action) associated with the image from which observations $s_{1:T}$ were obtained. We follow this approach for classification problems where we optimize the cross entropy loss to train the action network $f_a$ and backpropagate the gradients through the core and glimpse networks. The location network $f_l$ is always trained with REINFORCE.

## 4   Experiments

We evaluated our approach on several image classification tasks as well as a simple game. We first describe the design choices that were common to all our experiments:

**Retina and location encodings:** The retina encoding $\rho(x,l)$ extracts $k$ square patches centered at location $l$, with the first patch being $g_w \times g_w$ pixels in size, and each successive patch having twice the width of the previous. The $k$ patches are then all resized to $g_w \times g_w$ and concatenated. Glimpse locations $l$ were encoded as real-valued $(x,y)$ coordinates[2] with $(0,0)$ being the center of the image $x$ and $(-1,-1)$ being the top left corner of $x$.

**Glimpse network:** The glimpse network $f_g(x,l)$ had two fully connected layers. Let $Linear(x)$ denote a linear transformation of the vector $x$, i.e. $Linear(x) = Wx+b$ for some weight matrix $W$ and bias vector $b$, and let $Rect(x) = max(x,0)$ be the rectifier nonlinearity. The output $g$ of the glimpse network was defined as $g = Rect(Linear(h_g) + Linear(h_l))$ where $h_g = Rect(Linear(\rho(x,l)))$ and $h_l = Rect(Linear(l))$. The dimensionality of $h_g$ and $h_l$ was 128 while the dimensionality of $g$ was 256 for all attention models trained in this paper.

**Location network:** The policy for the locations $l$ was defined by a two-component Gaussian with a fixed variance. The location network outputs the mean of the location policy at time $t$ and is defined as $f_l(h) = Linear(h)$ where $h$ is the state of the core network/RNN.

| (a) 28x28 MNIST | |
| --- | --- |
| Model | Error |
| FC, 2 layers (256 hiddens each) | 1.69% |
| Convolutional, 2 layers | 1.21% |
| RAM, 2 glimpses, $8 \times 8$, 1 scale | 3.79% |
| RAM, 3 glimpses, $8 \times 8$, 1 scale | 1.51% |
| RAM, 4 glimpses, $8 \times 8$, 1 scale | 1.54% |
| RAM, 5 glimpses, $8 \times 8$, 1 scale | 1.34% |
| RAM, 6 glimpses, $8 \times 8$, 1 scale | 1.12% |
| RAM, 7 glimpses, $8 \times 8$, 1 scale | **1.07**% |

| (b) 60x60 Translated MNIST | |
| --- | --- |
| Model | Error |
| FC, 2 layers (64 hiddens each) | 6.42% |
| FC, 2 layers (256 hiddens each) | 2.63% |
| Convolutional, 2 layers | 1.62% |
| RAM, 4 glimpses, $12 \times 12$, 3 scales | 1.54% |
| RAM, 6 glimpses, $12 \times 12$, 3 scales | **1.22**% |
| RAM, 8 glimpses, $12 \times 12$, 3 scales | **1.2**% |

Table 1: Classification results on the MNIST and Translated MNIST datasets. FC denotes a fully-connected network with two layers of rectifier units. The convolutional network had one layer of 8 $10 \times 10$ filters with stride 5, followed by a fully connected layer with 256 units with rectifiers after each layer. Instances of the attention model are labeled with the number of glimpses, the number of scales in the retina, and the size of the retina.

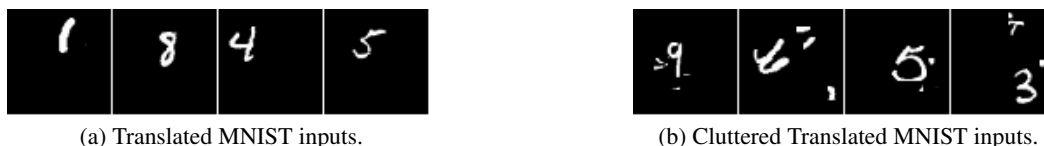

(a) Translated MNIST inputs.    (b) Cluttered Translated MNIST inputs.

Figure 2: Examples of test cases for the Translated and Cluttered Translated MNIST tasks.

**Core network:** For the classification experiments that follow the core $f_h$ was a network of rectifier units defined as $h_t = f_h(h_{t-1}) = Rect(Linear(h_{t-1}) + Linear(g_t))$. The experiment done on a dynamic environment used a core of LSTM units [10].

## 4.1 Image Classification

The attention network used in the following classification experiments made a classification decision only at the last timestep $t = N$. The action network $f_a$ was simply a linear softmax classifier defined as $f_a(h) = \exp(Linear(h))/Z$, where $Z$ is a normalizing constant. The RNN state vector $h$ had dimensionality 256. All methods were trained using stochastic gradient descent with minibatches of size 20 and momentum of 0.9. We annealed the learning rate linearly from its initial value to 0 over the course of training. Hyperparameters such as the initial learning rate and the variance of the location policy were selected using random search [3]. The reward at the last time step was 1 if the agent classified correctly and 0 otherwise. The rewards for all other timesteps were 0.

**Centered Digits:** We first tested the ability of our training method to learn successful glimpse policies by using it to train RAM models with up to 7 glimpses on the MNIST digits dataset. The "retina" for this experiment was simply an $8 \times 8$ patch, which is only big enough to capture a part of a digit, hence the experiment also tested the ability of RAM to combine information from multiple glimpses. We also trained standard feedforward and convolutional neural networks with two hidden layers as a baselines. The error rates achieved by the different models on the test set are shown in Table 1a. We see that the performance of RAM generally improves with more glimpses, and that it eventually outperforms a the baseline models trained on the full $28 \times 28$ centered digits. This demonstrates the model can successfully learn to combine information from multiple glimpses.

**Non-Centered Digits:** The second problem we considered was classifying non-centered digits. We created a new task called Translated MNIST, for which data was generated by placing an MNIST digit in a random location of a larger blank patch. Training cases were generated on the fly so the effective training set size was 50000 (the size of the MNIST training set) multiplied by the possible number of locations. Figure 2a contains a random sample of test cases for the 60 by 60 Translated MNIST task. Table 1b shows the results for several different models trained on the Translated MNIST task with 60 by 60 patches. In addition to RAM and two fully-connected networks we also trained a network with one convolutional layer of 16 $10 \times 10$ filters with stride 5 followed by a rectifier nonlinearity and then a fully-connected layer of 256 rectifier units. The convolutional network, the RAM networks, and the smaller fully connected model all had roughly the same number of parameters. Since the convolutional network has some degree of translation invariance built in, it

| (a) 60x60 Cluttered Translated MNIST | |
|---|---|
| Model | Error |
| FC, 2 layers (64 hiddens each) | 28.58% |
| FC, 2 layers (256 hiddens each) | 11.96% |
| Convolutional, 2 layers | 8.09% |
| RAM, 4 glimpses, $12 \times 12$, 3 scales | 4.96% |
| RAM, 6 glimpses, $12 \times 12$, 3 scales | 4.08% |
| RAM, 8 glimpses, $12 \times 12$, 3 scales | 4.04% |
| RAM, 8 random glimpses | 14.4% |

| (b) 100x100 Cluttered Translated MNIST | |
|---|---|
| Model | Error |
| Convolutional, 2 layers | 14.35% |
| RAM, 4 glimpses, $12 \times 12$, 4 scales | 9.41% |
| RAM, 6 glimpses, $12 \times 12$, 4 scales | 8.31% |
| RAM, 8 glimpses, $12 \times 12$, 4 scales | 8.11% |
| RAM, 8 random glimpses | 28.4% |

Table 2: Classification on the Cluttered Translated MNIST dataset. FC denotes a fully-connected network with two layers of rectifier units. The convolutional network had one layer of 8 $10 \times 10$ filters with stride 5, followed by a fully connected layer with 256 units in the $60 \times 60$ case and 86 units in the $100 \times 100$ case with rectifiers after each layer. Instances of the attention model are labeled with the number of glimpses, the size of the retina, and the number of scales in the retina. All models except for the big fully connected network had roughly the same number of parameters.

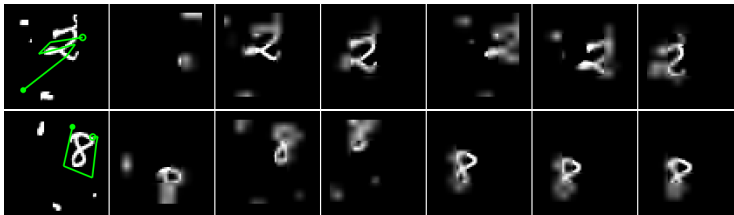

Figure 3: Examples of the learned policy on $60 \times 60$ cluttered-translated MNIST task. Column 1: The input image with glimpse path overlaid in green. Columns 2-7: The six glimpses the network chooses. The center of each image shows the full resolution glimpse, the outer low resolution areas are obtained by upscaling the low resolution glimpses back to full image size. The glimpse paths clearly show that the learned policy avoids computation in empty or noisy parts of the input space and directly explores the area around the object of interest.

attains a significantly lower error rate of $1.62\%$ than the fully connected networks. However, RAM with 4 glimpses gets slightly better performance than the convolutional network and outperforms it further for 6 and 8 glimpses, reaching $1.2\%$ error. This is possible because the attention model can focus its retina on the digit and hence learn a translation invariant policy. This experiment also shows that the attention model is able to successfully search for an object in a big image when the object is not centered.

**Cluttered Non-Centered Digits:** One of the most challenging aspects of classifying real-world images is the presence of a wide range clutter. Systems that operate on the entire image at full resolution are particularly susceptible to clutter and must learn to be invariant to it. One possible advantage of an attention mechanism is that it may make it easier to learn in the presence of clutter by focusing on the relevant part of the image and ignoring the irrelevant part. We test this hypothesis with several experiments on a new task we call Cluttered Translated MNIST. Data for this task was generated by first placing an MNIST digit in a random location of a larger blank image and then adding random 8 by 8 subpatches from other random MNIST digits to random locations of the image. The goal is to classify the complete digit present in the image. Figure 2b shows a random sample of test cases for the 60 by 60 Cluttered Translated MNIST task.

Table 2a shows the classification results for the models we trained on 60 by 60 Cluttered Translated MNIST with 4 pieces of clutter. The presence of clutter makes the task much more difficult but the performance of the attention model is affected less than the performance of the other models. RAM with 4 glimpses reaches $4.96\%$ error, which outperforms fully-connected models by a wide margin and the convolutional neural network by over $3\%$, and RAM trained with 6 and 8 glimpses achieves even lower error. Since RAM achieves larger relative error improvements over a convolutional network in the presence of clutter these results suggest the attention-based models may be better at dealing with clutter than convolutional networks because they can simply ignore it by not looking at it. Two samples of learned policy is shown in Figure 3 and more are included in the supplementary materials. The first column shows the original data point with the glimpse path overlaid. The

location of the first glimpse is marked with a filled circle and the location of the final glimpse is marked with an empty circle. The intermediate points on the path are traced with solid straight lines. Each consecutive image to the right shows a representation of the glimpse that the network sees. It can be seen that the learned policy can reliably find and explore around the object of interest while avoiding clutter at the same time. Finally, Table 2a also includes results for an 8-glimpse RAM model that selects glimpse locations uniformly at random. RAM models that learn the glimpse policy achieve much lower error rates even with half as many glimpses.

To further test this hypothesis we also performed experiments on 100 by 100 Cluttered Translated MNIST with 8 pieces of clutter. The test errors achieved by the models we compared are shown in Table 2b. The results show similar improvements of RAM over a convolutional network. It has to be noted that the overall capacity and the amount of computation of our model does not change from $60 \times 60$ images to $100 \times 100$, whereas the hidden layer of the convolutional network that is connected to the linear layer grows linearly with the number of pixels in the input.

### 4.2 Dynamic Environments

One appealing property of the recurrent attention model is that it can be applied to videos or interactive problems with a visual input just as easily as to static image tasks. We test the ability of our approach to learn a control policy in a dynamic visual environment while perceiving the environment through a bandwidth-limited retina by training it to play a simple game. The game is played on a 24 by 24 screen of binary pixels and involves two objects: a single pixel that represents a ball falling from the top of the screen while bouncing off the sides of the screen and a two-pixel paddle positioned at the bottom of the screen which the agent controls with the aim of catching the ball. When the falling pixel reaches the bottom of the screen the agent either gets a reward of 1 if the paddle overlaps with the ball and a reward of 0 otherwise. The game then restarts from the beginning.

We trained the recurrent attention model to play the game of "Catch" using only the final reward as input. The network had a 6 by 6 retina at three scales as its input, which means that the agent had to capture the ball in the 6 by 6 highest resolution region in order to know its precise position. In addition to the two location actions, the attention model had three game actions (left, right, and do nothing) and the action network $f_a$ used a linear softmax to model a distribution over the game actions. We used a core network of 256 LSTM units.

We performed random search to find suitable hyper-parameters and trained each agent for 20 million frames. A video of the best agent, which catches the ball roughly $85\%$ of the time, can be downloaded from `http://www.cs.toronto.edu/~vmnih/docs/attention.mov`. The video shows that the recurrent attention model learned to play the game by tracking the ball near the bottom of the screen. Since the agent was not in any way told to track the ball and was only rewarded for catching it, this result demonstrates the ability of the model to learn effective task-specific attention policies.

## 5 Discussion

This paper introduced a novel visual attention model that is formulated as a single recurrent neural network which takes a glimpse window as its input and uses the internal state of the network to select the next location to focus on as well as to generate control signals in a dynamic environment. Although the model is not differentiable, the proposed unified architecture is trained end-to-end from pixel inputs to actions using a policy gradient method. The model has several appealing properties. First, both the number of parameters and the amount of computation RAM performs can be controlled independently of the size of the input images. Second, the model is able to ignore clutter present in an image by centering its retina on the relevant regions. Our experiments show that RAM significantly outperforms a convolutional architecture with a comparable number of parameters on a cluttered object classification task. Additionally, the flexibility of our approach allows for a number of interesting extensions. For example, the network can be augmented with another action that allows it terminate at any time point and make a final classification decision. Our preliminary experiments show that this allows the network to learn to stop taking glimpses once it has enough information to make a confident classification. The network can also be allowed to control the scale at which the retina samples the image allowing it to fit objects of different size in the fixed size retina. In both cases, the extra actions can be simply added to the action network $f_a$ and trained using the policy gradient procedure we have described. Given the encouraging results achieved by RAM, applying the model to large scale object recognition and video classification is a natural direction for future work.

## Footnotes

[1]Depending on the scenario it may be more appropriate to consider a sum of *discounted* rewards, where rewards obtained in the distant future contribute less: $R = \sum_{t=1}^{T} \gamma^{t-1} r_t$. In this case we can have $T \to \infty$.

[2]We also experimented with using a discrete representation for the locations $l$ but found that it was difficult to learn policies over more than 25 possible discrete locations.

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
