[Supplementary Material]

# Supplementary Material

## Dynamic Environment Model Details

The architecture of the network used in the dynamic environment experiments in Section 4.2 was as follows. The retina received three patches centered around the chosen location at three scales. The first scale was a 6 by 6 patch sampled at full resolution, the second patch was 12 by 12 resized to 6 by 6f, and the third patch was 24 by 24 resized to 6 by 6. The connectivity of the glimpse network was the same as the one used in the classification experiments described in Section 4, i.e. two fully connected layers. The core network consisted of 256 LSTM cells. At each step, the outputs of the LSTM were fed into three output networks. The first was a linear layer outputting the $(x, y)$ of the next glimpse. The second was a linear layer with a single output that predicted the expected reward given the current state. The expected reward was used as a state-dependent baseline instead of the constant baseline used in the classification experiments. The third output layer was a linear layer with three outputs followed by a softmax nonlinearity. This softmax modeled the probabilities of the three possible actions (*left*,*right*, and *do nothing*). The input to the LSTM was a vector of zeros. Training was done by running the LSTM model until the ball reached the bottom and then updating the entire network using the algorithm described in Section 3.2.

## Policy Examples

Figure 1: Examples of the learned policy on $60 \times 60$ cluttered-translated MNIST task. Column 1: The input image from MNIST test set with glimpse path overlaid in green (correctly classified) or red (false classified). Columns 2-7: The six glimpses the network chooses. The center of each image shows the full resolution glimpse, the outer low resolution areas are obtained by upscaling the low resolution glimpses back to full image size. The glimpse paths clearly show that the learned policy avoids computation in empty or noisy parts of the input space and directly explores the area around the object of interest.

Figure 2: Examples of the learned policy on $60 \times 60$ cluttered-translated MNIST task. Column 1: The input image from MNIST test set with glimpse path overlaid in green (correctly classified) or red (false classified). Columns 2-7: The six glimpses the network chooses. The center of each image shows the full resolution glimpse, the outer low resolution areas are obtained by upscaling the low resolution glimpses back to full image size. The glimpse paths clearly show that the learned policy avoids computation in empty or noisy parts of the input space and directly explores the area around the object of interest.

Figure 3: Examples of the learned policy on $60 \times 60$ cluttered-translated MNIST task. Column 1: The input image from MNIST test set with glimpse path overlaid in green (correctly classified) or red (false classified). Columns 2-7: The six glimpses the network chooses. The center of each image shows the full resolution glimpse, the outer low resolution areas are obtained by upscaling the low resolution glimpses back to full image size. The glimpse paths clearly show that the learned policy avoids computation in empty or noisy parts of the input space and directly explores the area around the object of interest.