[Reviews · NeurIPS 2014]

Submitted by Assigned_Reviewer_3

An attentional mechanism is added to a feedforward convolutional
network, with recurrence introduced by the feedback of succeeding
attentional focus locations. The system integrates information across
many loops of this process, yielding both a classification and a
series of "fixation" points. The system is trained using RL, due to
issues of differentiability making the performance gradient
unavailable. The system achieves competitive performance, with a
dramatic reduction in computational burden.

This work tackles an important problem, and makes good progress. To
my mind, the main weakness is the RL-based training. It seems to me
that, with some machinations, it should be possible to render the
system differentiable. But this is a rather minor criticism: really
it is a suggestion for follow-on work. The fact that this approach
works so well *despite* a learning algorithm based on RL and therefore
necessarily slower than gradient-based methods speaks in *favour* of
the network structure presented here, as this is evidence that it
easier to train than I would have expected.
Summary: A convolutional network, augmented with a focus-of-attention spotlight,
an iterative mechanism for shifting this spotlight, and a way to
integrate information over time, trained with RL, exhibits good
performance, reduced computational burden, and generates sensible
fixation sequences.

Submitted by Assigned_Reviewer_26

The paper describes a method for sequentially learning sequences of action for classification and regression tasks in the context of visual scenes. The motivation comes from attentional models in vision where decisions are made by considering sequentially selected information. The model is formulated as a POMDP and the algorithm is implemented using a recurrent neural network. The hidden layer of this network serves as the state of the POMDP and the approximated policy is learned via the neural network. Experiments are performed on image classification tasks (noisy digits) and on toy control problem.
The paper is well motivated and clear. Some precisions could be added especially for the dynamic environment setting (both for the model, e.g. LSTM units and for the experiments). The paper originality, in the context of this application, is in the use of a recurrent net for dynamically updating the state of the model in its hidden layer and in the regression model for the prediction of the next place to look for.
The test beds on the other hand concern only very simple problems (both classification and dynamic prediction), so that the state space is probably not that big for all these problems. Experiments on stronger benchmarks are needed in order to effectively assess the capacity of this method to handle more realistic problems. Also, the proposed system could be compared with other sequential approaches. You could at least compare with a random policy + see the references below. The complexity of the method should be discussed, there is no mention of this aspect, and it should be compared to the complexity of other methods, both sequential and non-sequential. There is neither discussion nor analysis on the usefulness of the hidden layer recurrence. Did you made experiments with different memory sizes or decay rate (weights of the recurrent connections) or did you attempt to learn the weight values?
Detailed comments:
It seems that for the classification, the number of steps is fixed in advance and is the same for all the images. This is probably a useless assumption (by the way, how do you set up the value of this variable?).

Below are some references which could be interesting for this work.
Karayev, Sergey, Baumgartner, Tobias, Fritz, Mario, and Darrell, Trevor. Timely object recognition. In NIPS, pp. 1–9, 2
Cuccu, Giuseppe, Luciw, Matthew, Schmidhuber, Jurgen, and Gomez, Faustino. Intrinsically motivated neuroevolution for vision-based reinforcement learning. In Development and Learning (ICDL), 2011 IEEE International Conference on, volume 2, pp. 17. IEEE, 2011.012.
Contardo, Gabriella , Denoyer, Ludovic , Artieres, Thierry, Gallinari, Patrick, Learning State Representations in POMDP, ICLR 2014
Dulac-Arnold, Gabriel , Denoyer, Ludovic, Thome, Nicolas, Cord, Matthieu, Gallinari, Patrick, Sequentially Generated Instance-Dependent Image Representations for Classification, ICLR 2014
Summary: The paper proposes to use sequential learning model for vision tasks, when the environment is only partially known. The evaluation is not that strong: being performed on simple classification tasks and on a toy control problems, it does not show that the model is able to address real problems.

Submitted by Assigned_Reviewer_32

This paper proposes a recurrent neural network model inspired by the human perception system, called Recurrent Attention Model (RAM). This model is capable of extract information from image by adaptively selecting regions that are potentially interesting (by using the internal state of the network). This approach allows the amount of computation to be controlled independently of the input size. The parameters of the network are learned by maximizing the reward the agent can expect when interacting with the environment (using the REINFORCE rule). The model is validated on MNIST data. The two main advantages of the model are: (i) contrary to convolutional neural networks, the number of computation performed can be controlled independent of the input image size (in the CNN case, the number of computation increases linearly with the size of the input) and (ii) the model can ignore clutters present in images by considering only the relevant regions.

The authors propose a very original approach for image classification task, but unfortunately only show results on the MNIST dataset. The paper is in general very well written and provides a joyful reading. However, the results shown in the the paper are a bit unsatisfactory.

First of all, both the fully-connected (FC) and the convolutional (CNN) networks results from Table 1 are not representative. As can be seen in http://yann.lecun.com/exdb/mnist/ , both architectures can achieve better results than those proposed on Table 1 (results that are, indeed, better than the best result proposed by the authors). For this reason, the results for FC/CNN models in Table 2 (with the cluttered MNIST dataset) are not representative either.

Secondly, even though the authors state that the number of computation can be controlled (different from CNN), they do not demonstrate this advantage in the experimental results (a table comparing the time of inference in both models would be interesting). Besides, for such simple dataset as MNIST, both FC and CNN models achieve quite fast inference time (enough for real-time applications). Finally, the paper leaves the open question about how the system would perform in more real-world datasets (e.g. CIFAR-10/100, SVHN, Pascal, Imagenet)? It would be interesting to see how the model would work in such scenarios. The inference time in these cases could indeed be a big advantage if compared to classical CNNs.

[update] The idea of the paper is very original and surely a very promising approach to reduce computation in object recognition tasks. However, I feel the experimental section is missing empirical evidence that the model work in any realistic scenario. Also, the authors could have put more effort on the benchmark models (namely, fully-connected and convolutional neural networks models).
Summary: Interesting new algorithm explained on a very well written paper. However, some experimental results are missing. More importantly, the baseline results (FC and CNN) could be much improved and are not realistic represented on the paper.
Author Feedback
Author rebuttal: We thank all reviewers for their helpful and positive comments and suggestions.

Reviewers 1 and 3:

Results on Large-Scale Image Datasets: To our knowledge the presented method is the first successful end-to-end algorithm that models the visual attention task in static and dynamic scenarios. It merges reinforcement learning with deep learning using recurrent neural networks. We believe the experiments presented in the paper provide a solid demonstration of the capabilities of the algorithm.

We definitely agree that applying the model to real image datasets is an important direction and are currently working on this - with promising preliminary results. Nevertheless, in light of the novelty of the model, we have focused on experiments that provide a thorough investigation of its properties in a controlled and interpretable setup. Our focus was (a) to demonstrate that our method can learn good attentional policies in both static and dynamic environments, and (b) to highlight important qualitative differences in its behavior compared to alternative architectures commonly used in the literature. For the comparisons we aimed to make them as fair as possible by matching different models e.g. in terms of the number of free parameters.

Computational complexity of the attention model: The paper already includes some discussion of the computational complexity (see line 63ff), but we agree that this could be made more explicit and we will do so in the final version. In particular, in both fully connected and convolutional neural networks the number of floating point operations has a quadratic dependence on the width of the input image. For our attention model, the number of floating point operations is quadratic in the width of the retina, which is much smaller than the width of the image. In practice, we found that the inference time for the CNN increases by a factor of 2.33 when going from 60x60 to 100x100 pixels, which is close to the expected quadratic increase of 2.78; in contrast, inference time for a 4 glimpse attention model grows only by a factor of 1.11. The small increase in the runtime of RAM is due to the cost of resizing larger images in our current unoptimized implementation and could be removed completely by only resizing cropped regions.

Reviewer 1 (Assigned_Reviewer_26):

Thanks for the pointers to additional literature. We will incorporate these in the final version.

Dynamic environment details - We will include a full description of the dynamic environment and the LSTM model used for the experiment in the supplementary materials.

Comparisons to other sequential approaches - As suggested, we trained our model with a uniform random glimpse selection policy on the 60x60 and 100x100 Cluttered MNIST datasets and found that this achieves error rates roughly 3 times worse than the results we report for RAM using a learned glimpse policy. More precisely, training a model with 8 random glimpses on 60x60 Cluttered MNIST achieves 14.4% error while training it on 100x100 Cluttered MNIST achieves 28.4%. These results give further evidence that our method for learning the glimpse policy is effective. We will add these results to the final version.

Hidden layer recurrence - Making the hidden layer recurrent is important because it allows the model to maintain information about previous fixations over time. The weight sharing across steps prevents the number of parameters from scaling linearly with the number of fixations. We learn the full recurrent weight matrix (and all other network parameters). Making the recurrent hidden layer bigger leads to better results on the tasks we considered but we did not include these results because our goal was to make the different architectures comparable by allowing for (roughly) the same number of parameters. We will add these details to the paper.

Number of steps - The number of steps does not need to be the same for every image. Our model can be augmented with an additional action that decides when it will stop taking glimpses and make a classification. We found that our training algorithm can successfully learn when to stop but did not include these experiments in the paper due to space constraints. We will include these results in the final version, possibly in the supplemental material.

Reviewer 3 (Assigned_Reviewer_32):

MNIST results - First, we stress that our baselines achieve the same or better errors than comparable models from (http://yann.lecun.com/exdb/mnist/index.html). The 1.35% error of the fully connected model in Table 1a is a strong baseline because it is better than the best fully connected neural network result from the above page that does not use unsupervised pretraining or data augmentation (3-layer NN, 500+300 HU, softmax, cross entropy, weight decay achieving 1.53%). When applying the convolutional network used in our paper to MNIST 28x28 we obtain an error an error of 1% which is on par with the 0.95% obtained by a 5 layer convolutional network (LeNet 5) from the above result page. We will add this result to Table 1a for clarification. In line with the general goal of our evaluation, we chose the baseline architectures in the paper to be representative of typical neural networks widely used in the literature and set the numbers of parameter such as to roughly match to those of our RAM models for a comparison on equal footing.
We also stress that the results in Table 1b are on the 60x60 Translated MNIST dataset which is much more challenging than standard 28x28 MNIST, explaining the higher errors obtained by all baselines. Furthermore, the results in Table 2 include noise in the image that is produced from cropped digit parts which makes the task even more challenging. With these modifications, the performance of all of the standard feedforward algorithms degrade significantly as opposed to the learned attention based model.